# The Diverse Roles of Monocytes in Cryptococcosis

**DOI:** 10.3390/jof6030111

**Published:** 2020-07-16

**Authors:** Man Shun Fu, Rebecca A. Drummond

**Affiliations:** Institute of Immunology & Immunotherapy, Institute of Microbiology & Infection, University of Birmingham, Birmingham B15 2TT, UK; m.s.fu@bham.ac.uk

**Keywords:** monocytes, *Cryptococcus neoformans*, monocyte subsets, inflammatory monocytes, trained immunity, single-cell RNA-seq, flow cytometry, mass cytometry

## Abstract

Monocytes are considered to play a central role in the pathogenesis of *Cryptococcus neoformans* infection. Monocytes and monocyte-derived macrophages and dendritic cells are key components for the control of infection, but paradoxically they can also contribute to detrimental host responses and may even support fungal proliferation and dissemination. Simultaneously, the *C. neoformans* polysaccharide capsule can impair the functions of monocytes. Although monocytes are often seen as simple precursor cells, they also function as independent immune effector cells. In this review, we summarize these monocyte-specific functions during cryptococcal infection and the influence of *C. neoformans* on monocyte responses. We also cover the most recent findings on the functional and phenotypic heterogeneity of monocytes and discuss how new advanced technologies provide a platform to address outstanding questions in the field.

## 1. Introduction

Monocytes are innate immune cells that play critical roles in immune surveillance and the host response against infectious diseases. They make up 10% of circulating leukocytes and provide surveillance in all organs under steady-state conditions, thus providing a quick response following exposure to pathogens [1,2,3]. Upon infection, monocytes infiltrate tissue and not only do they act as a precursor for populations of macrophages and dendritic cells [4,5], but they also have their own effector functions including phagocytosis, antigen-presentation, and cytokine production [6]. While the plasticity of monocytes has been well-recognized, advanced technologies such as single-cell RNA sequencing and mass cytometry have recently revealed previously under-appreciated levels of the complex heterogeneity of monocytes and their subset-specific functions. Recent studies have shown that monocyte subsets vary among patients with different disease conditions [7], indicating that an understanding of monocyte heterogeneity can lead to new insights into disease pathogenesis and the identification of novel biomarkers of disease for diagnostic purposes [8,9,10]. 

Monocytes play a crucial role in controlling many invasive fungal infections; monocyte-deficient mice are more susceptible to infections with *Candida albicans*, *Aspergillus fumigatus*, and *Histoplasma capsulatum* [11,12,13,14,15,16,17,18]. For example, monocytes protect against candidiasis and aspergillosis by producing inflammatory cytokines and engaging in crosstalk with other immune cells such as natural killer (NK) cells, which trigger neutrophil activation for effective fungal killing [12,14,19]. However, for some fungal infections, the role of monocytes is still poorly understood, with both protective and detrimental roles being observed. In the case of *Cryptococcus neoformans* infections, monocyte function and differentiation are thought to be intimately linked with the pathogenesis of this infection, yet many questions still remain as to how these myeloid cells respond to *Cryptococcus* yeast and the consequences of these interactions on the resulting immune response. 

*Cryptococcus neoformans* is a fatal fungal pathogen among immunocompromised patients, causing over 180,000 deaths worldwide annually [20]. Cryptococcal infection initiates by inhalation of spores or desiccated yeast cells, which are found ubiquitously in the environment from multiple sources, including soils with contamination of bird excrement and some tree species [21,22,23]. Primary pulmonary infection is generally asymptomatic in healthy individuals whose innate immune system controls the infection before the fungus can disseminate [24]. However, if the immune system is compromised (such as by human immunodeficiency virus (HIV) co-infection, iatrogenic immune-suppression, or other underlying health disorder), *C. neoformans* yeast can disseminate throughout the host via the bloodstream, largely settling in the central nervous system (CNS). Cryptococcal meningitis has a mortality rate ranging from 20% to 80%, despite the availability of antifungal treatments [25], and although it is most common in immunocompromised patients, it can also occur in immunocompetent hosts [26].

Monocytes show beneficial roles in the innate host defense against *C. neoformans* [18,27]. They can phagocytose and eliminate the fungus, present antigens to T-cells, and produce cytokines to mediate crosstalk with other innate and adaptive immune cells [28,29,30]. However, paradoxically, they are also detrimental to the host in certain situations [31,32]. *C. neoformans* is a facultative intracellular pathogen that can replicate inside monocytes/macrophages [33]. Thus, these cells can harbor the fungus and shield it from the immune system, and may even act as “Trojan horses”, assisting fungal dissemination and brain invasion [31,34,35,36,37,38]. Simultaneously, *C. neoformans* has developed strategies to influence monocyte functions and behavior, resulting in a complex crosstalk that is critically involved in mediating the outcome of infection [39,40,41,42,43,44,45,46,47]. Therefore, monocytes are ideal targets for immunomodulatory therapies, the development of which is a key goal for the future treatment of invasive fungal infections.

Since there are many reviews focused on the interactions between macrophages and *Cryptococcus*, we intend to provide a comprehensive summary of monocyte function during cryptococcal infection, as well as the effect of *Cryptococcus* on monocyte functions. In particular, we discuss the most recent advances in monocyte biology, specifically the discoveries of new subsets and their diverse functions, and how new advanced technologies could be applied to expand our knowledge of the pathogenesis of cryptococcosis.

## 2. An Overview of Monocyte Subsets

Monocytes are typically divided into two major subsets in most studies: classical, inflammatory monocytes and non-classical, patrolling monocytes (Figure 1). In humans, these subsets are defined by their expression of CD14 (co-receptor for lipopolysaccharide) and CD16 (Fc gamma receptor IIIa) [48,49], where CD14^++^ CD16^-^ monocytes are inflammatory and CD14^+^ CD16^++^ are the non-classical subset [1,50] (Figure 1). In mice, Ly6C^high^ monocytes are the inflammatory subset while Ly6C^low^ monocytes are the patrolling subset [1,28,51]. Classical monocytes account for the majority of circulating monocytes, up to 90%, and are generally short-lived cells surviving for only 1 day [52,53]. These mobilize immediately into infected or injured sites and are responsible for diverse functions such as infection control, regulation of inflammation, and tissue repair. Upon entering tissue, inflammatory monocytes may also differentiate into dendritic cells or M1 or M2 macrophages depending on environmental cues [54]. For example, bacterial lipopolysaccharide (LPS) and cytokine interferon gamma (IFN-γ) drive the differentiation of M1 macrophages, which are defined by a high production of nitric oxide and pro-inflammatory cytokines [55,56,57,58]. M2 macrophages, on the other hand, are defined by arginase-1 expression and a wound-healing anti-inflammatory phenotype [55].

Non-classical monocytes are less well understood. They constantly patrol the circulation to engage in tissue repair and remove damaged or dead cells under homeostatic conditions and survive for up to 2 days in mice and 7 days in humans [52,53,59]. A third monocyte subset, which is less well characterized, has also been described. Intermediate monocytes (defined as CD14^++^ CD16^+^ in humans and Ly6C^int^ CD43^+^ in mice) have similar functions during an inflammatory response as classical monocytes, but also share some characteristics with non-classical monocytes, such as expression of some chemokine receptors [48,50,60] (Figure 1).

## 3. Monocyte Function during Cryptococcal Infection

Evidence from several clinical studies indicates that monocytes are required for optimal control of cryptococcosis in humans. Cohort studies on patients with HIV-associated cryptococcal meningitis showed that early mortality associated with a high fungal burden in the cerebrospinal fluid (CSF) is strongly associated with monocyte dysfunction, including reduced phagocytosis, reduced superoxide anion production, decreased expression of major histocompatibility complex (MHC) II cell surface receptor HLA-DR, and a lower production of inflammatory cytokine tumor necrosis factor alpha (TNF-α) [61,62]. Furthermore, a study on an immunocompetent patient with pulmonary cryptococcosis showed that although the patient had no apparent immune disorder, monocytes isolated from the peripheral blood manifested a significant reduction in killing activity against *C. neoformans*, as well as a significant impairment in TNF-α, interleukin 1 beta (IL-1β), and nitric oxide production [63]. Similarly, an immunological study on non-HIV cryptococcal patients with severe CNS disease found a predominance of M2 monocytes/macrophages in the CSF, which had a low level of phagocytosis and reduced TNF-α and IL-12 production [64].

During *C. neoformans* infection, monocyte subsets are recruited in an organ-specific fashion. Inflammatory monocytes are predominant (~90%) in the blood of patients with *C. neoformans* infection while intermediate and non-classical monocytes account for 8% and 4%, respectively [65]. Ly6C^high^ inflammatory monocytes rapidly infiltrate the murine lung upon exposure to *C. neoformans*, arriving as early as day 3 post-infection and peaking around day 14 [32]. This significant and sustained accumulation of classical monocytes is not the result of intrinsic proliferation, but rather the continuous CC chemokine receptor 2 (CCR2)-dependent recruitment from the bone marrow and spleen [32]. In contrast to the lung, Ly6C^low^ monocytes are predominantly recruited into the brain vasculature via the postcapillary venules (and subsequently the brain parenchyma) during *C. neoformans* infection, outnumbering the recruited Ly6C^high^ monocytes [38]. The recruitment of Ly6C^low^ monocytes is specific to the brain, since the frequency of this particular subset in the infected liver and kidney are similar to that of the blood [38]. Monocyte rolling and adhering to blood vessels within the brain is mediated by the interaction between vascular cell adhesion molecule 1 (VCAM1) on the brain endothelium and its ligand VLA4 (very late activation antigen-4) expressed by both Ly6C^low^ and Ly6C^high^ monocytes [38]. Expression of VLA4 is regulated by tumor necrosis factor receptor (TNFR) signaling, since *Tnfr^-/-^* mice had reduced numbers of both monocyte subsets in the infected brain, while the adoptive transfer of mixed monocytes isolated from wild-type and *Tnfr^-/-^* mice showed that TNFR-deficient monocytes are recruited less than wild-type monocytes to the brain during *C. neoformans* infection [38]. In humans, distinct subsets of monocytes appear to be associated with specific disease states. A study of non-HIV cryptococcal meningitis showed a predominant tissue infiltration of CD200R+ (a marker of the M2 phenotype) monocytes/macrophages in the CSF [64,66]. In cryptococcosis-associated immune reconstitution inflammatory syndrome (C-IRIS), a clinical condition that develops in a large number of AIDS (acquired immune deficiency syndrome) patients following antiretroviral therapy, there is an increased frequency of monocyte subsets with expression of activated monocyte markers (programmed death-ligand PD-L1 and alpha chain of interleukin-2 receptor CD25), implying that activation of monocytes may play a role in the development of C-IRIS [65].

Upon recruitment into infected tissues, monocytes play various roles in anti-cryptococcal defense. Both monocyte-derived macrophages and monocyte-derived dendritic cells (MoDCs) are crucial for the development of a protective type 1 response, characterized by the differentiation of M1 macrophages and IFN-γ production by CD4 T-cells [28,67]. The accumulation of M1 macrophages and MoDCs is strongly associated with an increase in monocytes within the lung, one of the main sites of cryptococcal infection [11,17,27]. *Ccr2^-/-^* mice, which have impaired migration of monocytes from the bone marrow, are highly susceptible to chronic *C. neoformans* infection. *Ccr2^-/-^* mice have an associated decreased recruitment of macrophages and MoDCs and a resulting higher fungal burden compared to wild-type mice [17,18]. However, it should be noted that impaired monocyte recruitment may actually be protective during acute infection, as we discuss in detail below.

In addition to acting as precursor cells, monocytes can also be effector immune cells in their own right. For example, in vitro studies have shown that monocytes engulf *C. neoformans* and can act as antigen-presenting cells for T-cells [30]. Monocytes express high levels of CD40 following exposure to *C. neoformans*, and the CD40-dependent interaction between monocytes and activated T-cells helps to regulate the secretion of pro-inflammatory cytokines such as IL-12, IFN-γ, TNFα, and IL-1β, leading to the activation of Th1 T-cells and the reciprocal enhancement of the fungicidal ability of monocytes [29,30]. Indeed, in vivo studies utilizing the CCR2-deficient mouse model also indicate that monocytes are essential for regulating lymphocyte activity, since *Ccr2^-/-^* mice had reduced recruitment of CD4^+^ T-cells, CD8^+^ T-cells, natural killer (NK) cells, and innate lymphoid cells to the infected lungs [16,32,68]. Moreover, monocytes were also found to drive protective Th1 responses, since deletion of CCR2 also diminished IFN-γ and IL-17A production, but increased IL-4 and IL-5 thus driving a non-protective Th2 type immune response [16,17,18].

While there are clear protective functions of monocytes in the defense against *C. neoformans* in mice and humans, monocytes have also been shown to have detrimental roles to the host. During acute infection, impaired recruitment of inflammatory monocytes (using the CCR2-DTR transgenic mouse model in which diphtheria toxin receptor (DTR) is expressed under control of the CCR2 promoter) reduces fungal burden and dissemination, and thus improves survival of mice from 26 to 31.5 days [32]. Monocyte depletion also reduced Th2 cytokines IL-5 and C-C chemokine ligand 5 (CCL5), indicating that monocytes may facilitate the generation of harmful Th2 responses during the early stages of infection [32]. In fact, inflammatory monocytes significantly upregulate transcription of genes associated with the M2 macrophage phenotype, which support cryptococcal intracellular survival and persistence and provide less anti-cryptococcal activity than M1 macrophages [32,69,70]. It is not clear how *C. neoformans* affects the function of monocytes during acute cryptococcosis and how this relates to fungal dissemination. However, it is well known that *C. neoformans* can use monocytes or macrophages as “Trojan horses” for fungal dissemination to the CNS [31,34,35,36,37,38]. Early studies showed that mice injected with *C. neoformans*-infected bone marrow-derived monocytes have higher fungal burdens in the brain, spleen, and lung when compared to mice infected with free yeast [31]. Phagocyte depletion (using clodronate liposomes) during the late stages of infection leads to a significant reduction of the fungal burden in the brain, spleen, and lung [31]. These results provide evidence that monocytes could promote fungal dissemination. Indeed, live cell imaging experiments have demonstrated that infected monocytes can transport *C. neoformans* across a monolayer of human brain endothelial cells cultured in vitro [36]. More recently, the use of intravital microscopy in vivo by Sun and colleagues has helped visualize monocyte-mediated transfer of *C. neoformans* to the brain via the postcapillary venules in live animals [38].

## 4. Cryptococcal Influence on Monocyte Function

In order to survive in the host, *C. neoformans* uses several strategies to evade or impair host immunity. Unlike other common human fungal pathogens, *C. neoformans* is covered by a capsule composed of polysaccharide glucuronoxylomannan (GXM) and galactoxylomannans (GalXM), in addition to mannoproteins [71,72]. The capsule, which *C. neoformans* can shed extracellularly and intracellularly, is one of the major virulence factors that is able to influence host immunity [39]. Capsular component GXM causes downregulation of the expression of co-stimulatory molecules on monocytes, including B7-1, B7-2, and CD40, thus affecting subsequent interactions with T-cells and alterations in cytokine production [41,43,44]. For example, GXM drives monocyte production of IL-10 [41]. IL-10 is a potent biological inhibitor of IL-12 expression. Reduced IL-12 production results in the decreased secretion of proinflammatory cytokines IFN-γ, TNF-α, and IL-1β, which adversely affects the fungicidal activity of monocytes [41,42]. These immune events could reflect that the GXM of *C. neoformans* diminishes a protective Th1 type response, which is crucial for clearance by affecting the function of monocytes. Another capsular component, mannoprotein (low-molecular-weight fractions with 35.6 and 8.2 kDa), plays a different role as an immunopotentiating antigen that triggers early induction of IL-12 secretion by human monocytes, which is required for the activation of IFN-γ+ T-cells [42]. However, it is likely that mannoprotein is often masked by capsular polysaccharides or competes with these molecules when interacting with immune receptors on monocytes. Although most capsular products are typically considered to be immunosuppressive, it has been shown that capsular products (including GXM, GalXM, and mannoprotein) can also induce monocyte-derived IL-6 production [45]; IL-6 is protective during cryptococcosis, but has also been shown to stimulate HIV replication in monocytes, thus the role of cryptococcal-derived products on the host immune response is complex, and the underlying immune status of the patient may significantly alter how these products affect monocyte function in vivo [45,73].

Another immunomodulatory cryptococcal factor that affects the function of monocytes is F-box protein Fbp1, a subunit of the SCF^Fbo1^ E3 ligase complex for protein ubiquitination and degradation [46]. Fbp1 was recently characterized as a new virulence factor, since mice infected with an *fbp1*-deficient mutant have increased recruitment of inflammatory monocytes to the lung, leading to an enhanced Th1- and Th17-type immune response and a concomitant reduction in pulmonary fungal burdens. Depletion of CCR2^+^ cells, which includes the inflammatory monocytes, in mice infected with the *fbp1* mutant resulted in abrogation of protective immunity and rapid death of the mice compared to animals infected with wild-type *C. neoformans*. The increased monocyte recruitment was associated with higher production of CCR2 ligands including CCL2, CCL7, and CCL12. While the exact mechanisms mediating anti-Fbp1 immunity are still unclear, it is hypothesized that deletion of Fbp1 may change the structure of cryptococcal cell wall, allowing better innate recognition by phagocytes and exposure of immunostimulatory mannoproteins, thus driving production of inflammatory chemokines. On the other hand, a fungal inositol polyphosphate kinase, Arg1, affects the cell surface architecture and secretion profile of *C. neoformans,* thus decreasing the recognition and phagocytosis of *C. neoformans* by host peripheral blood monocytes [47].

## 5. Monocyte-Mediated Immune Memory: Trained Immunity

The adaptive immune system has been traditionally considered to exclusively retain immunological memory to foreign antigens. However, in recent years there have been a growing number of studies demonstrating that monocytes and other innate immune cells can also develop immunological memory, which is termed trained immunity [74,75]. Trained cells undergo long-term functional reprogramming initiated by a primary stimulus, allowing them to give stronger and more rapid responses, and hence greater protection against secondary infections, even if they return to a non-activated state after the primary challenge. Numerous studies on trained immunity triggered by fungal pathogens, in particular *C. albicans*, have now been described. Monocytes undergo a series of histone modifications and metabolic changes when exposed to fungal cell wall β-glucan in vivo, which enhances their proinflammatory cytokine production and fungicidal activity upon infection with a lethal dose of *C. albicans*, ultimately leading to better survival of the infected mice [76,77]. Similar results are also obtained when mice are pre-treated with heat-killed *C. albicans* or have a primary infection with a low non-lethal dose of *C. albicans* [76]. Interestingly, trained monocytes also provide stronger proinflammatory responses upon secondary stimulation with diverse microbial antigens and pathogens, including LPS and *Staphylococcus aureus* [78,79]. Likewise, *Saccharomyces cerevisiae* β-glucan, chitin, chitosan, and acidic mannans can also trigger innate immune memory [80,81,82]. Pre-treatment with β-glucan has been shown to directly affect the modulation of hematopoietic stem and progenitor cells in the bone marrow, inducing expansion of progenitors of the myeloid lineage [83]. This effect contributes to elevated IL-1β production during a secondary challenge with LPS, leading to a therapeutic effect that lasts several weeks. Therefore, trained immunity can last for several weeks despite the short lifespan of circulating monocytes. While we have begun to better define trained immunity and the mechanisms governing this phenomenon, relatively few studies have been done that specifically analyze the development of trained immunity following exposure to *C. neoformans*, which may be important since it is thought that latent infection with *C. neoformans* may contribute toward the development of C-IRIS, and that low, frequent exposure to *C. neoformans* occurs in up to 70% of the population in some parts of the world [84]. Moreover, it has been shown that dendritic cells (DCs) develop memory-like cytokine responses upon challenge with an avirulent strain of *C. neoformans,* which is controlled by histone modifications [85]. Interestingly, these DCs fail to exhibit a significant cytokine response to secondary non-cryptococcal challenges, including LPS, *C. albicans,* or *S. aureus*, suggesting that DC memory resulting from stimulation with cryptococcal components is specific, but the component(s) remain to be identified. Whether similar phenotypes occur in *C. neoformans*-exposed monocytes has yet to be determined.

## 6. Functional and Phenotypic Heterogeneity of Monocyte Subsets

Monocytes appear to play diverse roles in anti-cryptococcal responses that can only be partially explained by timing and chronicity of infection. To help further understand these seemingly paradoxical functions, it will become necessary to analyze monocyte heterogeneity to determine whether specific functional monocyte subsets are responsible for protective versus detrimental functions. In recent years, new high-throughput technologies coupled with bioinformatics-based analysis have enabled immunologists to gain a deeper understanding of the cellular immune system. For example, single-cell RNA sequencing (scRNAseq) has led to the identification of new myeloid cell subsets in different tissues, including nerve-associated macrophages [86], as well as subsets that develop as a result of the inflamed microenvironment [87]. In addition, mass cytometry is a technique that combines flow cytometry with mass spectrometry and allows analysis of up to 40 markers at a single-cell resolution in a single sample [88,89,90,91,92,93]. The conventional gating based on CD14/CD16 may inaccurately define monocyte subsets since this single pair of surface markers is continuously distributed. Therefore, novel technologies are useful for identifying additional cell surface markers and further definition of monocyte subsets in mice and humans, as well as identifying entirely new monocyte subsets. For example, recent studies by Hoffmann et al. aimed to characterize the CD-ome of human monocytes and profiled a large panel of cell surface markers and revealed new surface markers that are unique to the classical (BLTR1, CD35, CD38, CD49e, CD89, CD96), intermediate (CD39, CD275, CD305, CDw328), and nonclassical (CD29, CD132) subsets, helping to improve the monocyte subset classification [94].

While flow cytometry and mass cytometry require preselection of markers and thus have a level of bias, single-cell RNA-seq, which allows transcriptional profiling of individual cells, offers an unbiased study of immune cell diversity. Novel pathogenic drivers and biomarkers can also be identified through the analysis of gene signatures within identified subsets under specific disease states, thus accelerating identification of therapeutic and diagnostic targets. For example, a recent study identified eight different monocyte subsets within the inflamed CNS in a mouse model of multiple sclerosis, in which defined transcriptional signatures suggests distinct functions for each subset [95]. The authors defined Cxcl10^+^ and Saa3^+^ monocyte subsets, which were independent of the Ly6C^high^ subset and caused direct tissue damage within the CNS. Depletion of these subsets alleviated symptoms and, thus, proved to be a potentially useful new therapeutic target. Cxcl10 production is also found to be elevated in the brain of *C. neoformans*-infected mice and is associated with the development of a protective immune response again *C. neoformans* [96,97]. However, high concentrations of Cxcl10 may also promote immunopathology and mortality by attracting large numbers of T-cells to the CNS, which is associated with the development of C-IRIS [98,99].

In other studies, characterization of monocyte subsets has led to the identification of novel biomarkers of disease. Single-cell RNA seq analysis of peripheral blood mononuclear cells (PBMCs) from septic patients identified 6 monocyte subsets (CD14^+^ NEAT1^+^ SELL^+^ monocyte, CD14^+^ NEAT1^+^ CCR1^+^ monocyte, NEAT1^+^ CD163^+^ monocyte, CCL3L1^+^ monocyte, CD16^+^ monocyte, monocytes/macrophage), of which the NEAT1^high^ monocytes were significantly expanded specifically during sepsis, indicating that high expression of NEAT1 can potentially be used as a diagnostic biomarker of sepsis [100]. Furthermore, in a study of human cytomegalovirus (HCMV) infection, two cell surface markers, MHCII and its chaperon CD74, were identified in latently infected monocytes [101]. Their expression level was inversely correlated with viral transcript levels, and CD74^high^ cells express a reduced immune-responsive gene signature, showing that HCMV drives monocytes into a state characterized by anergic-like phenotypes. Studies identifying additional phenotypes and/or analyzing gene expression of monocyte subsets during *C. neoformans* infection are still in their infancy. However, this could lead to the discovery of novel monocyte-expressed biomarkers that indicate the infectious stages and development of the host immune response, thus helping to determine precise treatment strategies.

## 7. Conclusions and Future Perspectives

The study of mechanisms driving monocyte heterogeneity is ongoing, and it will be worth investigating how this will shape the immune response against fungal pathogens. The functional and phenotypic characteristics of monocyte subsets can be significantly shifted by stimuli from the microenvironment. For example, delta-like 1 (Dll1) signals from endothelial cells interact with the NOTCH2 (neurogenic locus notch homolog protein 2) receptor on classical monocytes, leading to high expression of transcription factor NR4A1 (nuclear receptor subfamily 4 group A member 1) and the conversion from a classical to a non-classical phenotype [102,103]. The variety of monocyte subsets also depends on monocyte developmental origins. Monocytes arise from the myeloid lineage of the hematopoietic system and are generated continuously throughout life in the bone marrow from two different precursors, either a granulocyte–monocyte progenitor (GMP) or a monocyte–dendritic cell progenitor (MDP). When progenitors detect specific cytokines, a particular production program can be initiated to enhance numbers of functionally distinct monocytes accordingly [104]. For example, in response to *Toxoplasma gondii* infection, IFN-γ produced by NK cells in the bone marrow controls the transcriptional program of monocyte progenitors to produce MHCII^+^ Sca-1^+^ CX3CR1^-^ Ly6C^++^ monocytes [105], which have antimicrobial functions such as increased prostaglandin E_2_ production, an important lipid mediator for proinflammatory immune responses, prior to bone marrow egress [106]. During *C. albicans* infection in mice, bone marrow remodeling causes an increase in the generation of Ly6C^high^ monocytes [105], while in vitro studies have indicated that *C. albicans* can also drive the differentiation of common myeloid progenitors (CMPs) and GMPs into mature monocytes with a greater potential for phagocytosis and TNF-α and IL-6 production, mediated by TLR2/MyD88 (toll-Like Receptor 2/myeloid differentiation primary response 88) signaling [107]. How monocyte development and heterogeneity are influenced by cryptococcal infection is still poorly understood. This will be important to unravel since the interaction between monocytes and *C. neoformans* significantly influences the outcome of cryptococcosis through various functions that orchestrate both innate and adaptive antifungal immunity. The plasticity of monocytes is clearly beneficial for controlling the infection, but at the same time, it can be exploited by the fungus to benefit its survival and persistence within the host. Developing a deeper understanding of the heterogeneity and plasticity of monocytes during cryptococcosis will help to develop new strategies for adjunctive immune-based antifungal therapy. Indeed, this kind of monocyte-targeted therapeutic strategy has already been successfully used to treat cancer patients. A small-molecule CCR2 inhibitor, which prevents the recruitment of inflammatory monocytes, was given to patients with pancreatic cancer in a clinical trial. This study showed that the reduction of monocyte recruitment to pancreatic tumors helped decrease the number of immunosuppressive tumor-associated macrophages, leading to 49% of treated patients displaying reduced tumor size and 97% patients achieving tumor control [108]. Future studies employing newly available technologies have the advantage of identifying novel myeloid cell subsets that interact with *C. neoformans* during infection. These novels approaches have the potential to better understand the pathogenesis of cryptococcosis, which is critically needed to develop immune-based therapies for this fatal fungal infection.

## Figures and Tables

**Figure 1 jof-06-00111-f001:**
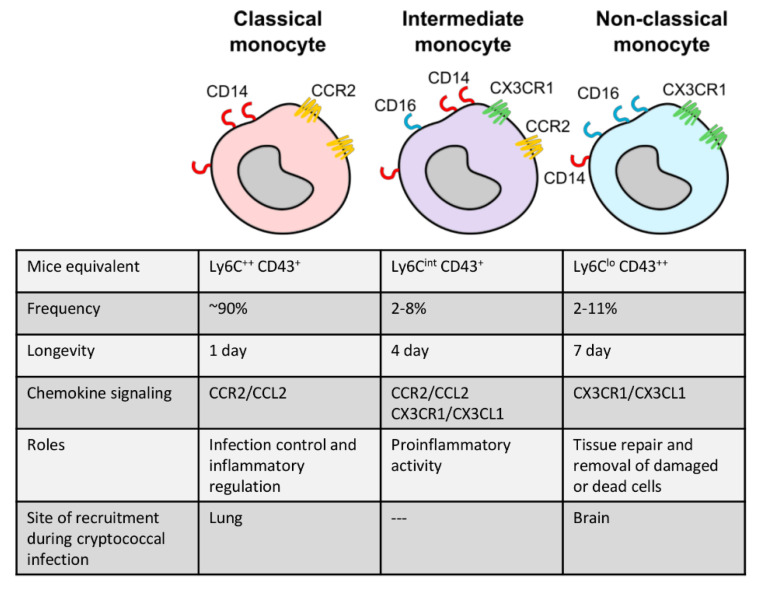
The three subtypes of monocytes that are best characterized to date: classical monocytes (CD14^++^ CD16^−^), intermediate monocytes (CD14^++^ CD16^+^), and non-classical monocytes (CD14^+^ CD16^++^). Murine markers and selected features for each subset are shown.

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
