# Peer review of "The Diverse Roles of Monocytes in Cryptococcosis"

_jof, 2020, doi:10.3390/jof6030111_

Round 1
Reviewer 1 Report
Overall the manuscript is well-written. There are a few suggestions for improvement.
Major comments:
- The title of the review is a bit redundant- “Cryptococcosis” is already the name of the infection caused by the fungus. “Cryptococcosis” is also not generally italicized or capitalized.
- In lines 34-36, it is a little confusing to list Cryptococcus neoformans as one of the infections to which monocyte-deficient mice are more susceptible - while this is true in some models of the infection, in other models monocytes can be detrimental (which the authors state later in the paper). In the infection model used in reference 37, deficiency of monocytes in both CCR2-DTR and Ccr2-/- mice renders them more resistant to infection. Thus, the sentences in lines 92-95 and lines 109-111 should also be restated/clarified.
- Lines 46-48- while bird excrement is a well-recognized reservoir of Cryptococcus, it is not the sole source. May want to note association with decaying matter including trees.
- Sections 4 and 5 seem somewhat disconnected from the rest of the paper. Could consider moving these sections closer to section 2, as these sections all introduce innate immunity to Cryptococcus while section 3 addresses the second goal of the paper- to discuss monocyte heterogeneity and new techniques to evaluate monocytes. Alternatively, may want to discuss how topics in sections 4 and 5 relate conceptually to the topics in section 3 or could be aided by research techniques discussed in section 3.
Minor comments:
- Typo line 25 sentence “Upon infection…”
- Add reference(s) for line 101 “Indeed, in vivo…” and line 165 “Expression of VLA4…”
- Check if correct reference line 163 “Monocyte rolling…”
- Run on sentences- may be helpful to break these down for better comprehension: line 50 “However, if…”, line 215 “Studies identifying…”, line 347 “Future studies…”,
- Figure 1: “remove” should probably be “removal of”?
Reviewer 2 Report
This review is providing a comprehensive and up to date summary and critique of the diverse roles of monocytes in Cryptococcosis, which will be a major interest to the readership of J Fungi, and more generally.
Minor comments and questions:
There are a few typographical errors throughout the manuscript.
Lines 46-48: 9 references are not required to make the point about environmental niches of cryptococcus etc. I suggest reducing substantially.
Lines 50-55: Dissemination of infection from lung to CNS occurs relatively commonly in patients without evidence of immunocompromise; it is true that presentation with CNS disease is proportionately more common in patients with immunocompromise.
Line 56: Use of the word “deviations” is incorrect; please replace
Lines 92-107: This section documents the importance of accumulation of M1 macrophages to controlling cryptococcal infection in the lung as indicated by findings in Ccr2-/- mice -which is in direct contrast to the findings in #37 (lines 108-113) whereby the opposite is true early in infection. The difference was explained by the authors of #37 by the fact that that the other studies addressed more subacute/chronic infection, (supported in their study by the observation that post-infection reduction of monocyte influx by use of diphtheria toxin did not impact mortality). This is an interesting observation requiring more study.
I suggest omitting #37 from line 92 and in lines 108-113 and indicating the actual reduction in survival (from 31.5 to 26 days).
Line 96-97: Please check that this reference (#55) is correct as it describes a case report of an apparently immunocompetent patient with pulmonary cryptococcosis. Some functional immune defects in PMN and monocytes were present while the patient was infected, but phagocytosis was not measured although growth inhibition was, and I couldn’t find a reference to APC functions in the manuscript. It would have been interesting to check if these abnormalities reversed after recovery to conclude that there was an underlying immune defect.
Line 165: Should ref #37 be ref #55??
In this section it would be important to indicate evidence that entry of infected monocytes into the CNS occurs at the level of post capillary venules (eg refs #80 and# 55), which had not been described previously.
Reviewer 3 Report
This review sets out to discuss the roles of monocytes in cryptococcal infection and pathogenesis. The introduction provides a good background within the field of fungal pathogenesis and underlines the paradoxical protective and deleterious roles of this group of cells. Section two provides a nice summary of known functions of monocytes during infection, focused mostly on the beneficial functions. The next section offers an introduction to the subtypes of monocytes and their characteristics, on the way to section 4, discussion how monocyte function is altered by cryptococcal infection. This section evokes more of the deleterious roles monocytes have. For the last section another dimension of role types played by monocytes is explored, sketching out the evidence for and significance of trained immunity.
Overall this is a nice bit of reviewing, which makes some of the immunology more accessible for those not immersed in that field. I have some relatively minor criticisms.
First, I felt confused at first about the distinction between monocytes and their derivatives (I’ve never seen the word “deviations” used that way), and where the focus would be. It is not until lines 68-69 that it was clarified for me that macrophages and DCs would not be in the focus, and an emphatic statement was reserved for lines 95-96:
“In addition to acting as precursor cells, monocytes can also be effector immune cells in their own right.”
Putting such a statement in the abstract or early in the introduction would have been very helpful.
A necessary difficulty with such a review the order in which to introduce different topics while maintaining a clear, thorough approach. Line 85 brings up M2 monocytes before the concept is fully introduced in the following section. Similarly, around line 110 the term “inflammatory monocytes” appears off and on, but its significance is lost until section 3. In both cases, the details and examples brought up might be better moved to later in the manuscript.
The rest of my comments are more minor:
Line 47: regarding inhalation of “encapsulated yeast cells.” The question of whether or not there is significant capsule on the infectious propagule has not been settled as far as I know. Please reference. I have often seen the term “desiccated” there. Early studies on the size of Cryptococcus particles in soil suggest that they take on a very different form from that seen in vivo. (See Neilson et al 1977 I&I 17(3) p.634)
After doing a nice job of introducing the marker sets for inflammatory and patrolling monocytes in humans and mice (including a very useful figure), lines 170 and 174 toss out two more surface marker phenotypes without laying down any significance for the markers. The former case was helpful for a reader that does not speak fluent CD-ese. The latter was thus confusing.
Lines 318-320: An interesting detail that might go better in Section 3 than in the conclusion.
Lines 347-350: perhaps more an editor’s job, but this final sentence needs to be 2 or 3. I could not digest it all at once.
Round 2
Reviewer 1 Report
The authors have addressed the previous major comments. The organization of the paper is more logical now. I believe the manuscript is appropriate for publication. I have the following minor comments:
-
- It may be helpful to mention that monocytes differentiate into dendritic cells in Section 2 so it is not as much of a surprise when it comes up in Section 3.
- Typo line 28 (previously line 25) sentence “Upon infection…” has not been corrected- "can" should be deleted.
- Lines 50-51 "environment/environmental" used twice
- Line 60- add subject to first phrase- "it is more common"
- Line 124 change to “account”
- Line 141 change to “macrophages”
- Line 151 run on sentence “Ccr2-/- mice…”
- Consistency in using “wild-type” vs “wildtype”
- Line 401 may want to include a noun “These <blank>” to aid comprehension
Author Response
All minor changes have now been made and are highlighted in yellow in the revised manuscript.